# Lactate-induced histone lactylation by p300 promotes osteoblast differentiation

Erika Minami[1,2], Kiyohito Sasa[1]*, Atsushi Yamada[1], Ryota Kawai[2], Hiroshi Yoshida[2], Haruhisa Nakano[2], Koutaro Maki[2], Ryutaro Kamijo[1†]

1 Department of Biochemistry, School of Dentistry, Showa University, Tokyo, Japan, 2 Department of Orthodontics, School of Dentistry, Showa University, Tokyo, Japan

† Deceased.
* k-sasa@dent.showa-u.ac.jp

**Data Availability Statement:** All relevant data are within the manuscript and its Supporting information files.

**Funding:** This work was supported by a Grant-In-Aid for Scientific Research (KAKENHI) from the

## Abstract

Lactate, which is synthesized as an end product by lactate dehydrogenase A (LDHA) from pyruvate during anaerobic glycolysis, has attracted attention for its energy metabolism and oxidant effects. A novel histone modification-mediated gene regulation mechanism termed lactylation by lactate was recently discovered. The present study examined the involvement of histone lactylation in undifferentiated cells that underwent differentiation into osteoblasts. C2C12 cells cultured in medium with a high glucose content (4500 mg/L) showed increases in marker genes (*Runx2, Sp7, Tnap*) indicating BMP-2-induced osteoblast differentiation and ALP staining activity, as well as histone lactylation as compared to those cultured in medium with a low glucose content (900 mg/L). Furthermore, C2C12 cells stimulated with the LDH inhibitor oxamate had reduced levels of BMP-2-induced osteoblast differentiation and histone lactylation, while addition of lactate to C2C12 cells cultured in low glucose medium resulted in partial restoration of osteoblast differentiation and histone lactylation. These results indicate that lactate synthesized by LDHA during glucose metabolism is important for osteoblast differentiation of C2C12 cells induced by BMP-2. Additionally, silencing of p300, a possible modifier of histone lactylation, also inhibited osteoblast differentiation and reduced histone lactylation. Together, these findings suggest a role of histone lactylation in promotion of undifferentiated cells to undergo differentiation into osteoblasts.

## Introduction

Production of lactate, a metabolite of anaerobic glycolysis, increases when demand for oxygen and ATP exceeds cellular supply, such as with exercise or infection [1]. It was previously considered to be a by-product, though is now known to be used intracellularly or released into adjacent organs as an energy source [2, 3]. Lactate has also been found to promote osteoblast differentiation via stabilization of oxia-inducible factor-1α (HIF-1α) and GPR81-PKC-Akt signaling [4, 5], while a recent study presented findings showing that lactylation by lactate is caused by a modification of histones and involved in epigenetics [6].

The term epigenetics refers to regulation of gene expression without alteration of gene sequence [7]. Typical epigenetic modifications include DNA methylation and histone

Japan Society for the Promotion of Science (21K16936, 23K09128). The funders had no role in study design, data collection and analysis, decision to publish, or preparation of the manuscript.

**Competing interests:** NO authors have competing interests.

modifications, with the former involving binding of methyl groups to DNA and regulation of gene expression [8–10]. On the other hand, histones are proteins that bind to DNA, and histone modifications play important roles in regulating changes in the higher-order structure of chromosomes and maintenance of cell fate through various chemical modifications, such as phosphorylation, acetylation, methylation, and ubiquitination, as well as others [11–14].

Histones are important for epigenetics, in which gene expression is regulated by acquired modifications of chromatin, and recently lactylation by lactate was found to be a novel modification of histones [6]. Lactylation is a modification in which a substructure of lactyl CoA binds to the lysine of histone proteins. As for the role of histone lactylation, when macrophages become polarized to M1 (inflammatory type) due to bacterial infection, the glycolytic system is stimulated, and lactate production and lactylation of lysine residues in histones are increased. This results in induction of gene expression necessary for polarization to M2, which is responsible for the convergence of inflammation [6]. Others have reported that the tumor metabolite lactate promotes tumorigenesis by modulating MOESIN lactation and enhancing TGF-β signaling in regulatory T cells [15] and that histone lactylation contributes to tumorigenesis by facilitating YTHDF2 expression for development of ocular melanoma [16]. In other words, lactate is not only a byproduct of glycolysis, but also an important molecule that regulates gene expression through histone modification.

In our previous study, small interfering RNA (siRNA)-mediated knockdown of monocarboxylate transporter 1 (MCT), a transporter of monocarboxylates (lactate, pyruvate, ketone bodies, etc.), was found to suppress osteoblast differentiation in the myoblast cell line C2C12 via activation of p53 [17]. It was also shown that MCT inhibition and knockdown inhibited osteoclast differentiation and function. *Mct1* knockdown in macrophages promoted RANKL-induced osteoclastogenesis, whereas *Mct2* knockdown inhibited it. In mature osteoclasts, *Mct2* knockdown reduced the number of osteoclasts and suppressed bone resorption. Together, these results indicate that MCT1 is a negative regulator and MCT2 a positive regulator of osteoclast differentiation, as previously suggested [18]. Thus, it is considered that intracellular and extracellular transport of monocarboxylates, typified by lactate, by MCTs are key factors for differentiation of osteoblasts and osteoclasts, which are important for bone remodeling. However, the detailed mechanism of gene regulation by monocarboxylate transport for osteoblast differentiation remains largely unknown. Based on results of the present study, it is suggested that lactate production by glucose metabolism regulates osteoblast differentiation through epigenetics other than energy metabolism.

## Results

### High glucose level in C2C12 cells promoted osteoblast differentiation and increased histone lactylation

C2C12 cells are commonly cultured in medium with a high concentration of glucose (4500 mg/L) for maintenance and differentiation. BMP-2 stimulation inhibits C2C12 cells from differentiating into myotube cells and induces their conversion into osteoblast-like cells [19, 20]. First, to confirm changes in osteoblast differentiation due to differences in glucose metabolism, C2C12 cells were placed into high or low glucose (900 mg/L) medium, and osteoblast differentiation was induced by BMP-2. The results showed that high glucose increased both intracellular and extracellular lactate levels as compared to low glucose (Fig 1a). Furthermore, the number of alkaline phosphatase (ALP)-positive cells was increased and ALP activity enhanced when cultured in high glucose as compared to low glucose medium (Fig 1b). Interestingly, C2C12 cells cultured in high glucose medium showed upregulated expression of the transcription factors Runx2 (*Runx2*) and Osterix (*Sp7*), which are required for osteoblast

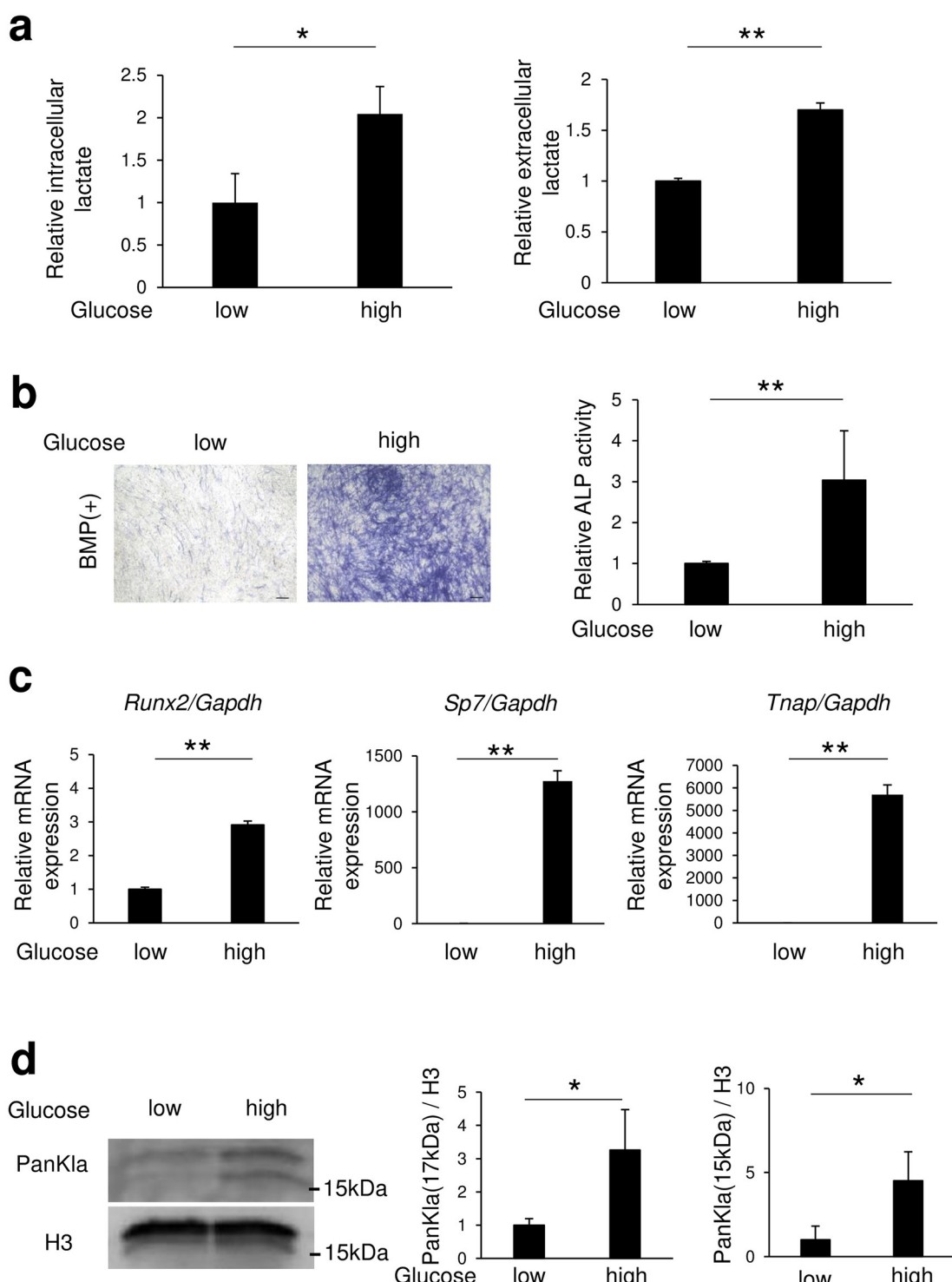

**Fig 1. High glucose concentration promoted osteoblast differentiation and increased histone lactylation levels in C2C12 cells.** C2C12 cells were incubated for 72 (a right, b, c) or 24 (a left, d) hours after addition of BMP-2 (150 ng/ml) into medium with a high (4500 mg/L) or low (900 mg/L) glucose concentration. Amounts of (a) lactate, and (b) ALP staining and activity are shown. (c) Osteoblast differentiation marker genes were determined using real-time RT-PCR. (d) Expression of lactyl lysine after addition of BMP-2 was evaluated by western blotting. Data are expressed as mean ± SD (n = 3–5). *, **Significantly different from control group (*p <0.05, **p <0.01).

differentiation, as well as of ALP (*Tnap*), a specific osteoblast differentiation gene (Fig 1c). Next, histone proteins from C2C12 cells cultured in high or low glucose medium were extracted, and histone lactylation was detected using an anti-lactyl-lysine antibody. The results showed that C2C12 cells cultured in high glucose medium had a greater level of histone lactylation (17 and 15 kDa bands) than those cultured in low glucose medium (Fig 1d).

## Inhibition of lactate dehydrogenase A caused inhibition of osteoblast differentiation and reduced histone lactylation

Lactate dehydrogenase A (LDHA) is an enzyme that converts pyruvate produced in the glycolytic system to lactate [21], while oxamidic acid (oxamate) is an LDHA inhibitor. Oxamate was added to medium at different concentrations (0, 5, 10, 20 mM), then lactate levels in cell lysate or medium used to culture C2C12 cells were measured. Both intracellular and extracellular concentrations of lactate were found to be decreased in an oxamate concentration-dependent manner (Fig 2a). Furthermore, oxamate decreased alkaline phosphatase (ALP)-positive cells and inhibited ALP activity in a concentration-dependent manner (Fig 2b), with *Sp7* and *Tnap* mRNA expressions in C2C12 cells were downregulated. However, *Runx2* mRNA expression was not changed in C2C12 cells with addition of oxamate (Fig 2c), whereas histone lactylation levels (17 and 15 kDa bands) were reduced (Fig 2d). In contrast, histone acetylation levels were unchanged by Oxamate (S1 Fig). Histone acetylation was not detected band of 15 kDa.

## Reduced lactate production by low glucose partially responsible for suppression of osteoblast differentiation

Next, whether lactate production is involved in osteoblast-like cell differentiation of C2C12 cells was examined by culturing in low glucose medium. First, lactate (10 or 20 mM) was added to cells cultured in low glucose medium with BMP-2, and the concentrations of intracellular and extracellular lactate in cell lysates and medium were measured(Fig 3a). Extracellular lactate in cultures of C2C12 cells cultured in low-glucose medium was increased by lactate addition, while the intracellular lactate concentration was also found to be partially recovered with addition of lactate as compared to cultures performed with high gulucose medium. Furthermore, suppression of BMP-2 introduced ALP activity in C2C12 cells induced by the low glucose medium was recovered by addition of lactate (Fig 3b). Additionally, the expression of *Runx2* mRNA was completely recovered when lactate was added to cultures with low glucose as compared to high glucose medium. While the expressions of *Sp7* and *Tnap* mRNA tended to upregulated when lactate was added as compared to low glucose medium only, those expression levels were lower than seen in high glucose medium (Fig 3c). Addition of lactate to low glucose medium also increased the histone lactylation levels (17 and 15 kDa bands) as compared to C2C12 cells cultured in low glucose medium, though those were lower than seen with high glucose medium (Fig 3d).

## Osteoblast differentiation by histone lactylation regulated by p300

Histone acetyltransferase (p300) has been reported as a possible enzyme for promoting the function of histone lactyl transferase [6, 22]. Introduction of *Ep300* siRNA into C2C12 cells resulted in knockdown of *Ep300* mRNA expression by approximately 75% (S2 Fig). In contrast, extracellular and intracellular lactate concentrations were not changed by *Ep300* siRNA (Fig 4a). We also found that ALP-positive cells were decreased and ALP activity was suppressed with use of *Ep300* siRNA (Fig 4b), while the expressions of *Sp7* and *Tnap* mRNAs were also decreased. Although the difference was not statistically significant, expression of

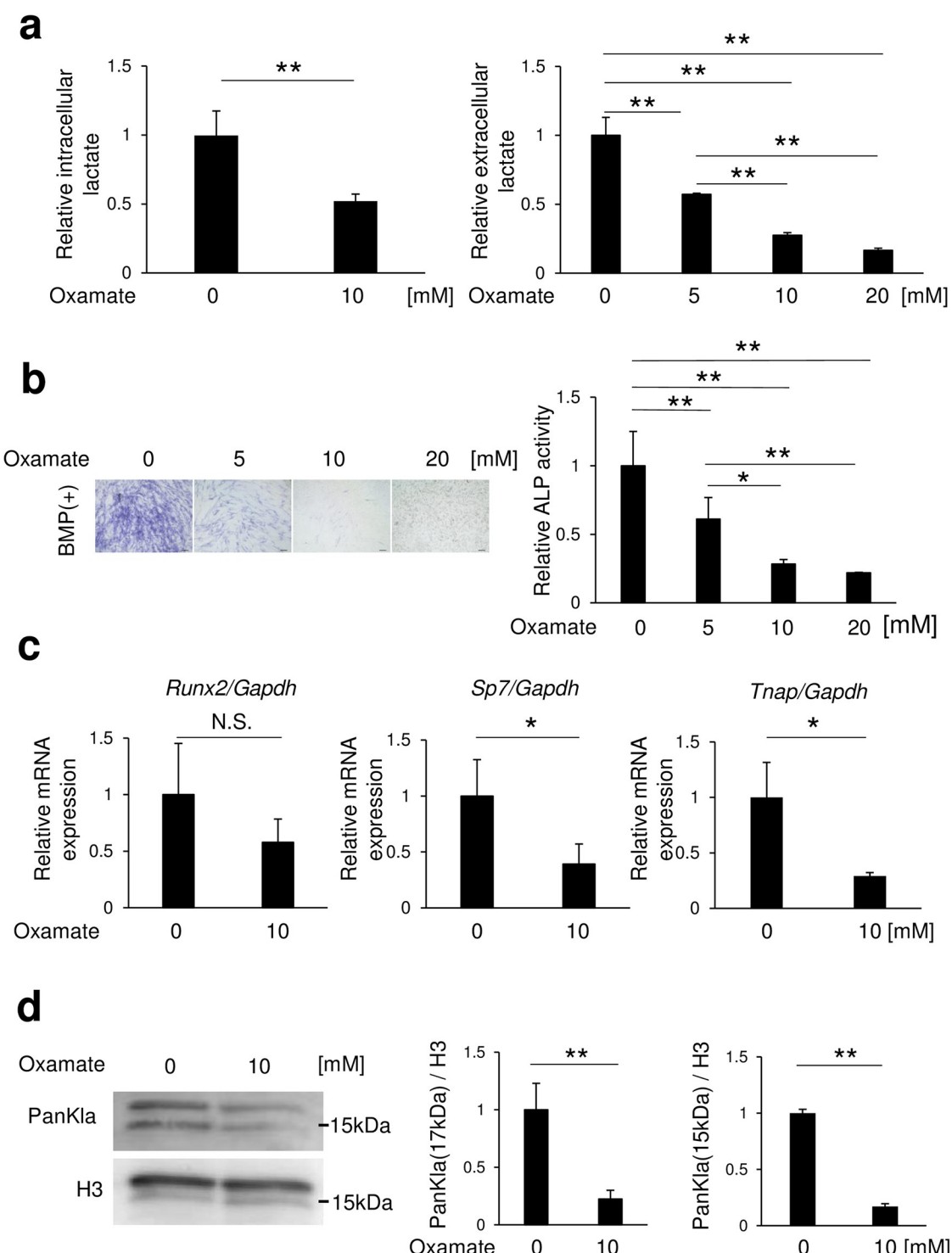

**Fig 2. Inhibition of LDHA reduced osteoblast differentiation and lactyl lysine levels.** C2C12 cells were incubated for 72 (a right, b, c) or 24 (a left, d) hours after addition of BMP-2 with oxamate. Amounts of (a) lactate, and (b) ALP staining and activity are shown. (c) Osteoblast differentiation marker genes were determined using real-time RT-PCR. (d) Expression of lactyl lysine after addition of BMP-2 and oxamate was evaluated by western blotting. Data are expressed as mean ± SD (n = 3). *, **Significantly different from control group (*p <0.05, **p <0.01). NS, not significant.

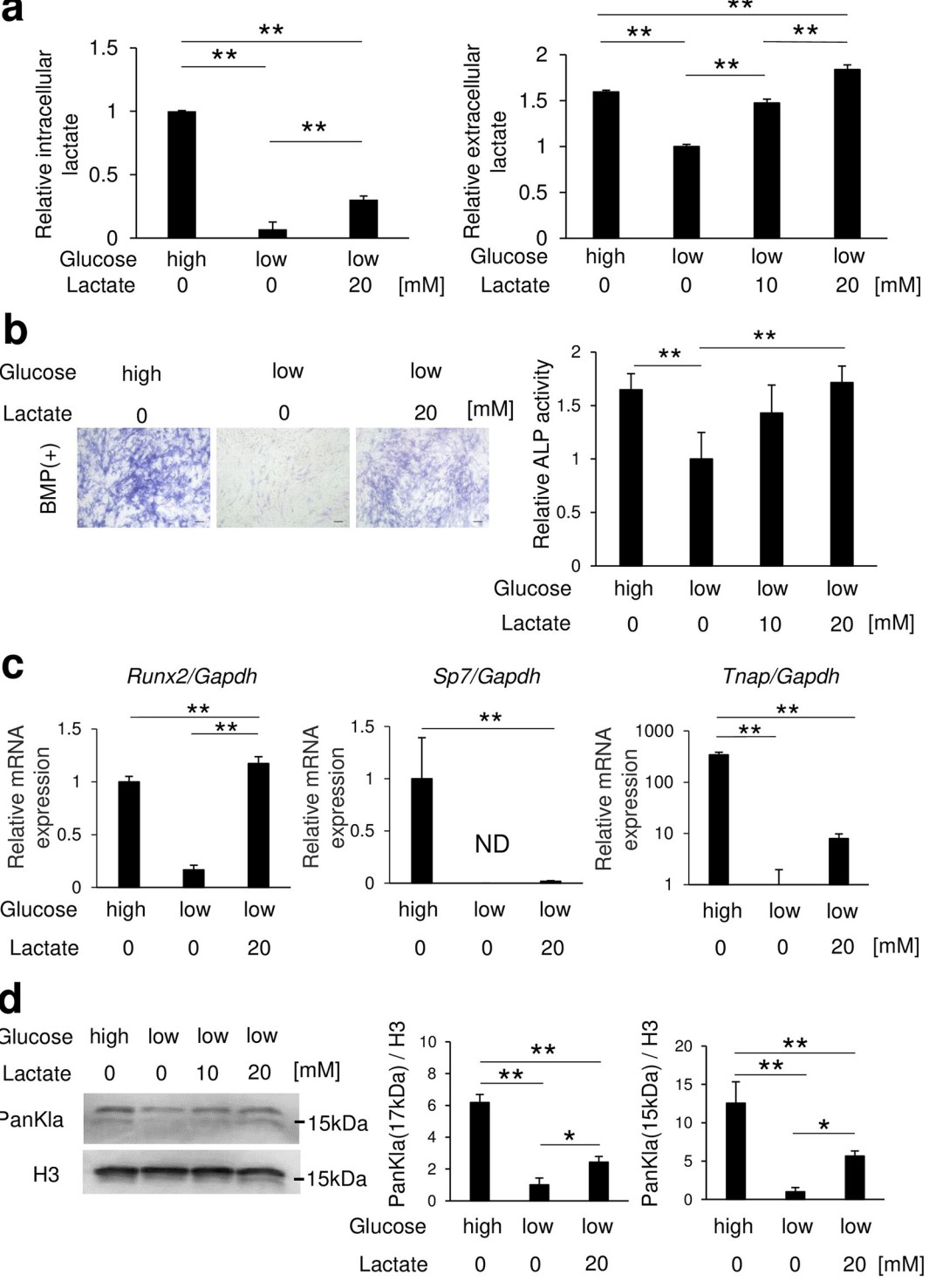

**Fig 3. Lactate recovered suppression of osteoblast differentiation and decreased histone lactylation levels caused by low glucose.** C2C12 cells were incubated for 72 (a right, b, c) or 24 (a left, d) hours after placing in medium with a high or low glucose concentration in the absence or presence of lactate. Amounts of (a) lactate, and (b) ALP staining and activity are shown. (c) Osteoblast differentiation marker genes were determined using real-time RT-PCR. (d) Expression of lactyl lysine under the same conditions was evaluated by western blotting. Data are expressed as mean ± SD (n = 3–4). *, **Significantly different from control group (*p <0.05, **p <0.01). ND: not detected.

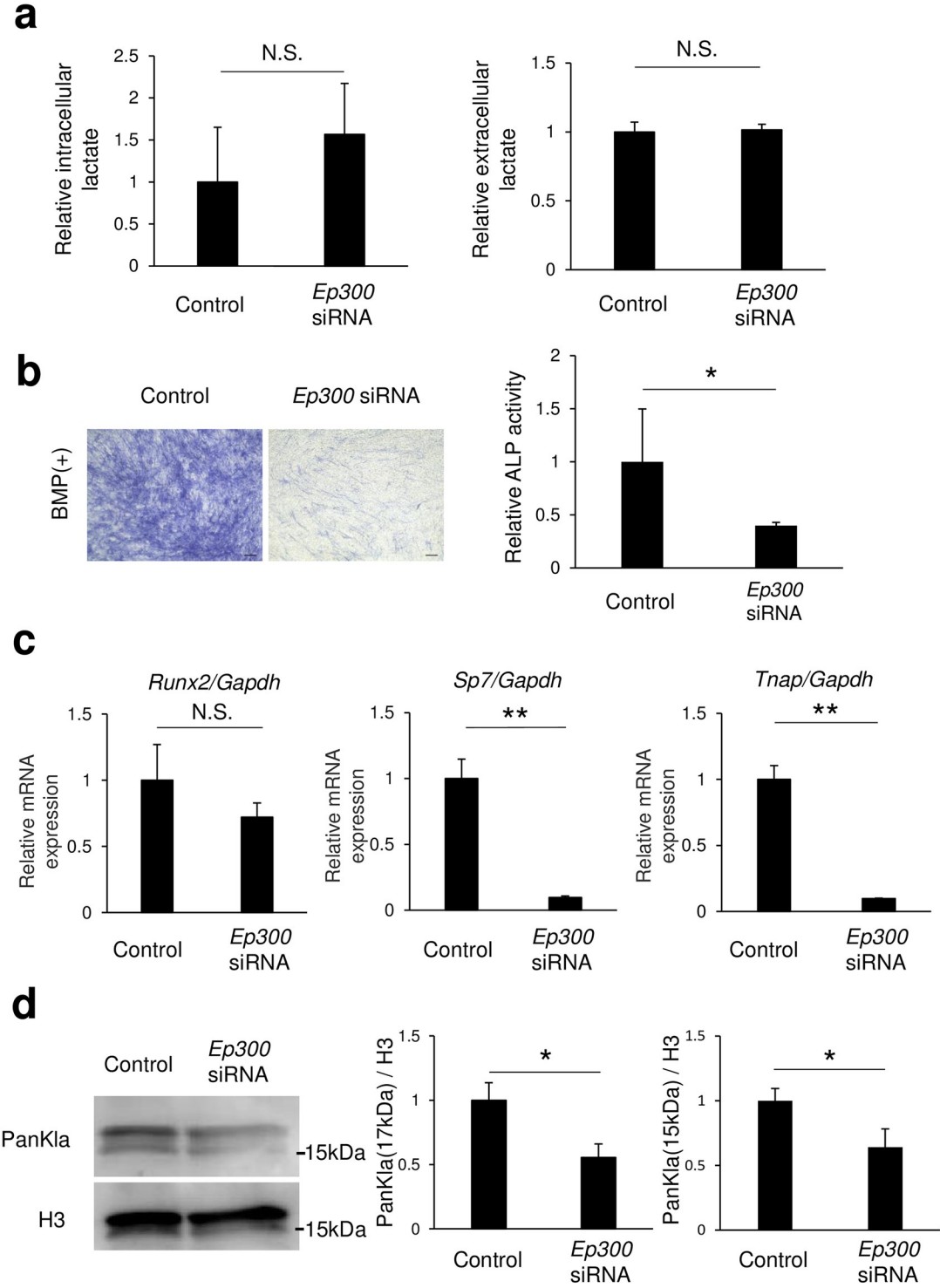

**Fig 4.** *Ep300* **siRNA suppressed osteoblast differentiation and decreased histone lactylation levels.** Following introduction of *Ep300* siRNA, C2C12 cells were cultured for 72 (a right, b, c) or 24 (a left, d) hours after addition of BMP-2 to the medium. Amounts of (a) lactate, and (b) ALP staining and activity are shown. (c) Osteoblast differentiation marker genes were determined using real-time RT-PCR. (d) Expression of lactyl lysine under the same conditions was evaluated by western blotting. Data are expressed as mean ± SD (n = 3–5). *, **Significantly different from control group (*p <0.05, **p <0.01). NS, not significant.

*Runx2* mRNA was decreased in C2C12 cells by *Ep300* siRNA (Fig 4c) and histone lactylation levels (17 and 15 kDa bands) were also reduced (Fig 4d). Additionally, histone acetylation levels were reduced by addition of *Ep300* siRNA (S3 Fig). Histone acetylation was not detected band of 15 kDa.

## Discussion

The present findings demonstrate that lactate produced by glucose metabolism promotes differentiation of cells to osteoblasts at an early stage by histone modification via p300. It is known that lactate promotes osteoblast differentiation by stabilization of hypoxia inducible factor 1a (HIF-1α) [4]. Another study that examined the pre-osteoblast cell line MC3T3-E1 found that PTH treatment-mediated osteoblast differentiation was promoted by positive feedback via GPR81-PKC-Akt signaling due to increased lactate production [5]. However, there are few reports regarding osteoblast differentiation via histone modification by lactate.

In the present study, undifferentiated C2C12 cells were used to examine changes in histone lactylation and mRNA expression during the early stage of differentiation. The results showed that increased glucose metabolism increased osteoblast differentiation and histone lactylation (Fig 1b–1d), while inhibition of LDHA decreased those (Fig 2b–2d), thus indicating that intracellular lactate production is required for osteoblast differentiation. Furthermore, osteoblast differentiation and histone lactylation, which are decreased with a low level of glucose, were partially restored by addition of lactate (Fig 3b–3d). Those results indicate that an environment that includes low glucose leads to a lower level of energy production due to reduced glycolysis, while lack of lactate production reduces osteoblast differentiation. Notably, lactate production by glucose metabolism was found to positively regulate osteoblast differentiation through p300-mediated histone lactylation (Fig 4b–4d).

Use of different glucose concentrations resulted in alterations of the mRNA expression of transcription factors such as *Runx2* and *Sp7*, important for determining osteoblast differentiation (Fig 1c). However, oxamate and p300, which regulate histone lactylation, were not found to be involved in changes in *Runx2* mRNA, though may be involved in *Sp7* changes (Figs 2c and 4c). It is thus speculated that alterations in *Runx2* mRNA caused by changes in glucose concentration involve actions of lactate other than lactate-induced histone lactylation, such as energy metabolism and oxidative actions. The present results indicate that histone lactylation regulates *Sp7* rather than *Runx2*. Although *Sp7* (Osterix) is a downstream gene of Runx2, a previous study found that Osterix$^{-/-}$ mice showed complete absence of osteoblasts, demonstrating Osterix as a transcription factor essential for osteoblast differentiation [23]. However, in the present experiments, when lactate was added to low glucose medium, *Runx2* mRNA levels recovered to the same level as seen in the high glucose medium, but the increases in *Sp7* and *Tnap* mRNA levels were small (Fig 3c). There are two possible reasons for this. First, a partial restoration of histone lactylation was noted when lactate was added to low glucose as compared to high glucose medium (Fig 3d). Second, the post-gene expression metabolism of *Runx2* may be involved. A prior report noted that glucose uptake by Glut1 promotes osteoblast differentiation by suppressing AMPK-dependent proteasomal degradation of Runx2 [24] and it is conceivable that the present results may have also been due to AMPK-dependent proteasomal degradation of Runx2 by use of low glucose medium. Therefore, it is conceivable that the mRNA expression levels of *Sp7* and *Tnap*, downstream genes of *Runx2*, were slightly increased by histone lactylation. Nevertheless, the detailed mechanism remains to be elucidated.

Since p300 is a histone acetyltransferase, inhibition of osteoblast differentiation by p300 knockdown may involve histone acetylation [25, 26]. *Ep300* siRNA was found to decrease histone lactylation and acetylation levels (Fig 4d, S3 Fig). On the other hand, oxamate decreased

only histone lactylation levels, while histone acetylation levels were not changed (Fig 2d, S1 Fig). In addition, osteoblast differentiation genes (*Sp7*, *Tnap*) were decreased in the presence of oxamate. Based on these findings, it is considered that lactate produced during the process of glucose metabolism regulates histone lactylation levels via p300 and promotes osteoblast differentiation. A recent report also noted that histone lactylation (H3K18lac) in MC3T3-E1 cells promotes osteoblast differentiation via regulation of *JunB* expression at the late differentiation stage [27], with the present findings providing support for those results.

In conclusion, the results of the present study show that lactate produced during glucose metabolism regulates osteoblast differentiation from undifferentiated cells through the epigenetics of histone modification.

## Materials and methods

### Reagents

Recombinant human BMP-2 was obtained from R&D Systems. D-glucose, L-lactic acid, and oxamate (LDHA inhibitor) were purchased from FUJIFILM Wako Pure Chemical Corporation. The PanKla antibody (PTM1401RM) and PanKac (PTM101) were obtained from PTM Biolabs and the histone H3 antibody (ab1791) from Abcam. siRNA used for knockdown of target mRNA (Silencer® Select siRNA) was obtained from Thermo Fisher Scientific, Inc.

### Cell culture

C2C12 cells were purchased from RIKEN BioResource Center (C2C12 #RCB0987) and cultured in Dulbecco's modified Eagle's medium (DMEM, FUJIFILM Wako Pure Chemical Corporation) supplemented with 15% fetal bovine serum (FBS) for cell proliferation. For osteoblast differentiation, C2C12 cells were stimulated with DMEM supplemented with 2.5% FBS and 150 ng/ml rhBMP-2.

### ALP activity staining

C2C12 cells were seeded into 96-well plates at $1 \times 10^4$ cells/well and cultured in the presence of BMP-2 for 72 hours. Transfection of the cells with siRNA was performed at the time of seeding into the plates. Oxamate was added and pretreatment performed for two to three hours before the start of differentiation, then the culture medium was replaced with differentiation medium with Oxamate included. Cells were fixed for 30 minutes in 4% paraformaldehyde and washed with PBS, then incubated for 30 minutes at 37˚C with 100 mmol/L Tris-HCl buffer (pH 8.5) containing 270 μmol/L naphthol AS-MX phosphate (Sigma-Aldrich) and 1.4 mmol/L Fast blue BB (Sigma-Aldrich). After washing with tap water, they were observed under a microscope. The scale bar shown in the figures represents 100 μm.

### Determination of ALP activity

The conditions for cell culturing were the same as for ALP activity staining, noted above. For determination of ALP activity, cells were washed with PBS and homogenized with 100 μL of NP-40 under sonication. After incubation at 37˚C for 30 minutes, absorbance was measured at 405 nm.

### Determination of extracellular and intracellular lactate

After 24 hours or 72 hours of culturing of C2C12 cells, cell lysate or medium supernatant samples were obtained and absorbance measured at a wavelength of 450 nm using a Lactate Assay Kit-WST (Dojindo), according to the manufacturer's instructions.

### Real-time RT-PCR

Total RNA was extracted from C2C12 cells using TRIzol reagent (Invitrogen), according to the manufacturer's instructions, then reverse transcription reactions were performed using ReverTra ACE RT qPCR master Mix (Toyobo). Quantitative real-time RT-PCR was performed using a TaqManTM Gene Expression Assay (*Gapdh*, Mm99999915_g1; *Tnap*, Mm00475834_m1; *Runx2*, Mm00501584_m1; *Sp7*, Mm04209856_m1; *Ep300*, Mm00625535_m1). Gene signals were normalized against the amplified signal of *Gapdh*.

### Western blot analysis

C2C12 cells were seeded into six-well plates at $5 \times 10^5$ cells/well and cultured in the presence of BMP-2 for 24 hours. Histone proteins were extracted from cells using an Epi Quik Total Histone Extraction Kit (Epigen Tek), according to the manufacturer's instructions. Histone proteins were subjected to SDS-PAGE (10% polyacrylamide gel) under a reducing condition. The samples were placed in Mini-PROTEAN TGX gel and electrophoresis was performed, then the proteins were transferred to PVDF membranes and treated with primary antibodies for one hour, followed by secondary antibodies against Rabbit IgG (GE Healthcare) for on hour. Immunoreactive bands were visualized by an enhanced chemiluminescence reaction using an ECL Prime Western Blot Detection System (GE Healthcare). The intensity of the chemiluminescent bands was quantitatively analyzed using Versa Doc 5000 MP (Bio-Rad Laboratories). The ratio of the histone band intensity of lactyl lysine (Pan Kla) at approximately 17 and 15 kDa to the histone protein band intensity of total histone H3 was calculated. The band at approximately 17 kDa contained histone H2A, H2B, and H3 proteins, while that at approximately 15 kDa contained histone H4 proteins.

### Statistical analysis

Values are expressed as the mean ± SD. Student's *t*-test was used for comparisons between two groups. One-way ANOVA with post-hoc Tukey test was performed for comparing results from three or more groups. P-values less than 0.05 were considered to be statistically significant. All statistical analyses were performed with EZR or Microsoft® Excel® 2019 MSO. In the graph shown in Fig 3c (*Sp7/Gapdh*), the low glucose group was ND, thus Student's t-test was used to compare the high glucose group with the low glucose with lactate added group.

### Supporting information

**S1 Fig. Inhibition of LDH was not changed histone acetylation levels in C2C12 cells.** C2C12 cells were incubated for 24 hours after addition of BMP-2 (150 ng/ml) with Oxamate. Amounts of expression of acetyl lysine after addition of BMP-2 and oxamate was evaluated by western blotting. Data are expressed as mean ± SD (n = 3). *, **Significantly different from control group (*p <0.05, **p <0.01).
(PDF)

**S2 Fig. *Ep300* siRNA decreased expression of *Ep300* mRNA in C2C12 cells.** Following introduction of Ep300 siRNA, C2C12 cells were cultured for 72 hours after addition of BMP-2 to the medium. Expression of mRNA for *Ep300* were analyzed by real-time PCR. Amplification signals from the gene was normalized that of *Gapdh*. Data are expressed as mean ± SD (n = 3). *, **Significantly different from control group (*p <0.05, **p <0.01).
(PDF)

**S3 Fig. *Ep300* siRNA decreased histone acetylation levels in C2C12 cells.** Following intro-duction of *Ep300* siRNA, C2C12 cells were cultured for 24 hours after addition of BMP-2 to the medium. Amounts of expression of acetyl lysine was evaluated by western blotting. Data are expressed as mean ± SD (n = 3). *, **Significantly different from control group (*p <0.05, **p <0.01).
(PDF)

**S1 Raw images.**
(PDF)

## Author Contributions

**Conceptualization:** Kiyohito Sasa.

**Formal analysis:** Erika Minami.

**Funding acquisition:** Kiyohito Sasa.

**Investigation:** Erika Minami, Kiyohito Sasa, Ryota Kawai, Hiroshi Yoshida.

**Project administration:** Haruhisa Nakano, Koutaro Maki, Ryutaro Kamijo.

**Supervision:** Ryutaro Kamijo.

**Validation:** Erika Minami.

**Writing – original draft:** Kiyohito Sasa, Atsushi Yamada.

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
