## [Decision Letter · Decision Letter 0]

20 Jul 2023

PONE-D-23-14523Lactate-induced histone lactylation by p300 promotes osteoblast differentiationPLOS ONE

Dear Dr. Sasa,

Thank you for submitting your manuscript to PLOS ONE. After careful consideration, we feel that it has merit but does not fully meet PLOS ONE’s publication criteria as it currently stands. Therefore, we invite you to submit a revised version of the manuscript that addresses the points raised during the review process.

We look forward to receiving your revised manuscript.

Kind regards,

Atsushi Asakura, Ph.D

Academic Editor

PLOS ONE

“This work was supported by a Grant-In-Aid for Scientific Research (KAKENHI) from the Japan Society for the Promotion of Science (21K16936, 23K09128).”

“This work was supported by a Grant-In-Aid for Scientific Research (KAKENHI) from the Japan Society for the Promotion of Science (21K16936, 23K09128).The funders had no role in study design, data collection and analysis, decision to publish, or preparation of the manuscript.”

7. In your Data Availability statement, you have not specified where the minimal data set underlying the results described in your manuscript can be found. PLOS defines a study's minimal data set as the underlying data used to reach the conclusions drawn in the manuscript and any additional data required to replicate the reported study findings in their entirety. All PLOS journals require that the minimal data set be made fully available. For more information about our data policy, please see http://journals.plos.org/plosone/s/data-availability.

Reviewers' comments:

Reviewer's Responses to Questions

**Comments to the Author**

1. Is the manuscript technically sound, and do the data support the conclusions?

Reviewer #1: Yes

Reviewer #2: Partly

Reviewer #3: Yes

2. Has the statistical analysis been performed appropriately and rigorously? 

Reviewer #1: Yes

Reviewer #2: Yes

Reviewer #3: No

3. Have the authors made all data underlying the findings in their manuscript fully available?

Reviewer #1: Yes

Reviewer #2: Yes

Reviewer #3: Yes

4. Is the manuscript presented in an intelligible fashion and written in standard English?

Reviewer #1: Yes

Reviewer #2: Yes

Reviewer #3: Yes

5. Review Comments to the Author

Reviewer #1: In this manuscript, the authors investigated the function and mechanism of histone lactylation on osteoblast differentiation. The results showed a role of histone lactylation in promotion of undifferentiated cells to osteoblast differentiation. The paper shows evidence that supports the conclusions, however, the following aspects should be addressed.

Major Concerns:

1. Figure1d and Figure 2d: the author only repeats once on KLA, it’s not enough in standard scientific article.

2. The author designed a LDH inhibitor (oxamate) group to study the function of LDHA on osteoblast differentiation, why not add a gene overexpression group?

3. There is no significant in results.

4. Figure 2 and 3a: Add relative intracellular lactate results.

Minor Concerns:

Figure 1b: Add scale in your picture. Check it through the MS.

Figure1d and Figure 2d: The KD of proteins should be added.

Reviewer #2: In this study, the authors demonstrated that lactate produced by glycolysis regulates osteoblast differentiation via histone H3 lactylation by promoting histone lactyl transferase activity by HAT/p300, which epigenetically enhances the expression of genes promoting osteoblast differentiation.

This is an interesting study, but some issues need to be clarified.

1. In Figures 1d through 4d, there are two bands stained with anti-lactyl lysine antibody (PanKla) in H3. What do these two bands represent and density ratio of which band was measured? In addition, there are no error bars for PanKla/H3 ratio. Because these are very important data to show H3 lactylation by lactate, these experiments need to be repeated and statistical significance should be presented.

2. In Figure 2, an LDHA inhibitor, oxamate, dose-dependently reduced extracellular lactate concentration and ALP activity, and 10 mM oxamate reduced lactyl lysine in H3 to a similar extent to that under low glucose (Figure 1d). However, Runx2 expression was inhibited only mildly and insignificantly, which is also the case in Figure 4c. Without change in Runx2 expression, how osteoblast differentiation was suppressed by oxamate?

3. In Figure 3, addition of lactate to low glucose medium recovered extracellular lactate concentration, ALP activity, PanKla/H3 ratio and Runx2 expression. In contrast, there was almost no change in Sp7 and Tnap expression. How ALP activity was increased despite very little change in Tnap expression, and what was the explanation for these discrepant results in Runx2, Sp7 and Tnap between Figures2 and 3? In Figures 2a and 3a, only extracellular lactate concentration was demonstrated. However, what is important is intracellular lactate concentration, and the authors should demonstrate intracellular lactate concentration.

4. In Figure 4, gene expression profiles of Runx2, Sp7 and Tnap were similar to those under oxamate treatment in Figure 2, and were different from those in Figure 3. These discrepant results under different treatments give readers a concern about the validity of the authors’ conclusion. In addition, knockdown of p300/HAT causes inhibition of histone acetylation and may change expression of various genes aside from change in histone lactylation.

5. It is worthwhile to refer to and discuss about the paper by Karsenty group (Wei J, et al. Cell 151: 1576-91, 2015) demonstrating that glucose uptake via Runx2-dependent expression of Glut1 is required and that Glut1 and Runx2 crosstalk determines osteoblast differentiation.

Reviewer #3: The study by Minami et al. describes the role of p300 lactylation as a mechanism associated with osteoblast differentiation. The authors conducted an experimental design using C2C12 cells to demonstrate the involvement of lactate in BMP-2-induced differentiation. While the experiments support this mechanism, there are some aspects that should be evaluated by the authors.

Considering that histone lactylation is key to understanding the regulatory mechanism in this article, it is noteworthy that only one result is shown in the bar graph of Figures 1, 2, 3, and 4 (d). It is suggested to include more replicates and display their dispersion as standard deviation (SD) along with statistical analysis. Were both bands considered for relative quantification since two bands are observed? Whatever the case may be, it is recommended to clarify this in the Materials and Methods section.

To demonstrate the role of LDHA and the conversion of pyruvate to lactate, oxamate was used. Was a cell viability analysis performed to rule out possible cytotoxic effects?

For the statistical analyses, the Student's t-test was used. However, in Figures 2 and 3, more than two experimental groups were used. In such cases, it is more appropriate to use analysis of variance (ANOVA) and post-hoc multiple comparisons tests. Additionally, significant differences are indicated in Figure 3C (Tnap/Gapdh) between low/0 and low/20 mM lactate, but the graph should be modified on its axis to visualize this unclear difference. Furthermore, in this figure, differences between high and low are shown for the Sp7/Gapdh gene, but no expression was detected in one group. It should be clearly stated which groups are being compared in this figure.

6. PLOS authors have the option to publish the peer review history of their article (what does this mean?). If published, this will include your full peer review and any attached files.

Reviewer #1: No

Reviewer #2: No

Reviewer #3: No

---

## [Author Response · Author response to Decision Letter 0]

27 Aug 2023

Reply to Reviewers

Revision Notes

We sincerely appreciate the helpful comments from the reviewers in regard to our study, which provided new insights. As a result, additional experiments were performed according to the editor’s and reviewers’ suggestions, and revisions have been made, resulting in great improvements to the manuscript.

Reviewer 1

The reviewer’s comments and suggestions are sincerely appreciated, and were helpful to improve our study. Please note our responses following.

Major Concerns:

Comment #1:

Figure1d and Figure 2d: the author only repeats once on KLA, it’s not enough in standard scientific article.

Our reply to comment #1

This suggestion is greatly appreciated. We carefully reperformed the experiments (n=3), and added the findings as Figures 1d, 2d, 3d, and 4d in the revised version. 

Comment #2: 

The author designed a LDH inhibitor (oxamate) group to study the function of LDHA on osteoblast differentiation, why not add a gene overexpression group?

Our reply to comment #2

Thank you for pointing this out. The experiments were performed by introducing plasmid DNA into C2C12 cells. The results showed that overexpression of LDHA in C2C12 cells did not promote osteoblastic differentiation. Previous studies have shown that LdhA is highly expressed in muscle tissues [Mishra et al., Cancers (Basel). 11(6): 750, 2019., Markert, et al., Science 11;189(4197):102-14, 1975.]. Since C2C12 cells are mouse myoblasts, it is possible that overexpression of LdhA had little effect on osteoblast differentiation due to the high expression level of LdhA mRNA (Appendix Fig. 1).

Comment #3:

There is no significant in results.

Our reply to comment #3

The authors are not sure if the reviewer is referring to the results shown in Figure 2c (Runx2/Gapdh) or those in Figure 4a, c (Runx2/Gapdh). With confirmation, we will do our best to respond to the comment.

Comment #4:

Figure 2 and 3a: Add relative intracellular lactate results.

Our reply to comment #4

We appreciate this helpful suggestion. Intracellular lactate graphs have been added to the revised version of the manuscript (Fig. 2a, 3a). The intracellular lactate concentration was found to be correlated with histone lactylation.

Minor Concerns:

Comment #1:

Figure 1b: Add scale in your picture. Check it through the MS.

Our reply to comment #1

A scale bar indicating 100 μm has been added the photographs have also been replaced (Fig. 1b, 2b, 3b, 4b). Related details are now described in the Materials and Methods section of the revised version (page 12, lines 278-279).

Comment #2:

Figure1d and Figure 2d: The KD of proteins should be added.

Our reply to comment #2

Protein kDa values have been added to the photographs (Fig. 1d, 2d, 3d, 4d). Related details are now described in the Materials and Methods section of the revised version (page 14, lines 311-315).

Reviewer 2

We greatly appreciate the reviewer’s comments and suggestions regarding our study. Please note our responses following. We consider that the manuscript has been significantly improved.

Comment #1:

In Figures 1d through 4d, there are two bands stained with anti-lactyl lysine antibody (PanKla) in H3. What do these two bands represent and density ratio of which band was measured? In addition, there are no error bars for PanKla/H3 ratio. Because these are very important data to show H3 lactylation by lactate, these experiments need to be repeated and statistical significance should be presented.

Our reply to comment #1

Thank you for pointing this out. The density ratio of the upper band is approximately 17 kDa and it contains histones H2A, H2B, and H3, while that of the lower band is approximately 15kDa and it contains H4. As suggested by the reviewer, additional experiments (n=3) were conducted, with the findings shown in a graph after separating the approximately 17 and 15 kDa bands. These results are described in the Materials and Methods section of the revised manuscript (page 14, lines 311-315).

Comment #2:

In Figure 2, an LDHA inhibitor, oxamate, dose-dependently reduced extracellular lactate concentration and ALP activity, and 10 mM oxamate reduced lactyl lysine in H3 to a similar extent to that under low glucose (Figure 1d). However, Runx2 expression was inhibited only mildly and insignificantly, which is also the case in Figure 4c. Without change in Runx2 expression, how osteoblast differentiation was suppressed by oxamate?

Our reply to comment #2

The authors appreciate the comments from the reviewer and this helpful question. There are two important transcription factors related to differentiation of mesenchymal stem cells into osteoblasts, Runx2 and Osterix (Sp7). Osterix is a downstream gene of Runx2, though a previous study found that Osterix-/- mice showed complete absence of osteoblasts, demonstrating that Osterix is a transcription factor essential for osteoblast differentiation [Nakashima, et al., Cell. 11;108(1):17-29, 2002.]. Based on these findings, we consider that a reduction of intracellular lactate by oxamate suppresses osteoblast differentiation through reduction of Osterix gene expression. Comments in this regard have been added to the revised Discussion section (page 10, lines 221-225).

Comment #3:

1) In Figure 3, addition of lactate to low glucose medium recovered extracellular lactate concentration, ALP activity, PanKla/H3 ratio and Runx2 expression. In contrast, there was almost no change in Sp7 and Tnap expression.

2) How ALP activity was increased despite very little change in Tnap expression, and what was the explanation for these discrepant results in Runx2, Sp7 and Tnap between Figures2 and 3?

3) In Figures 2a and 3a, only extracellular lactate concentration was demonstrated. However, what is important is intracellular lactate concentration, and the authors should demonstrate intracellular lactate concentration.

Our reply to comment #3

Thank you for these important suggestions and questions. 

3-1) As noted regarding the difference between ALP activity and Tnap gene expression, the data shown in the Tnap/Gapdh and ALP activity graphs do not correlate with those obtained on 72 hours of culture. We newly checked gene expression at 24 hours after differentiation (Appendix Fig. 2), which adding lactate to low glucose medium showed that the Sp7 and Tnap mRNA levels were nearly the same as those in high glucose medium, while the mRNA levels of Runx2 were the same in nearly all of the groups. It is possible that the ALP activity graph may reflect gene expression in the early phase of culturing, though we do not have details in that regard to show.

3-2) We concluded that histone lactylation regulates Sp7 (Osterix) rather than Runx2, because neither oxamate stimulation nor Ep300 siRNA altered the Runx2 gene (Fig. 2c, 4c). Thus, it is considered that factors other than histone lactylation are involved in the increase in the Runx2 gene seen with the addition of lactate to low glucose medium. For example, the presence of a Gαi protein coupled receptor (GPR81), a lactate receptor, present on the cell membrane. A previous report noted that lactate increased Runx2 gene expression via GPR81-Akt signaling in MC3T3-E1 cells [Yu Wu, et al., Biochem Biophys Res Commun. 5;503(2):737-743, 2018.]. However, in the present experiments, addition of lactate to low glucose medium resulted in a slight increase in expression levels of Sp7 and Tnap (Fig. 3c). There are two possible reasons for this. First, a partial restoration of histone lactylation was noted when lactate was added to the low glucose as compared to the high glucose medium (Fig. 3d). Second, the post-gene expression metabolism of Runx2 may be involved. These factors are presented in the paper noted in Comment 5 from the reviewer below. Those authors noted that glucose uptake by Glut1 promotes osteoblast differentiation by suppressing the AMPK-dependent proteasomal degradation of Runx2 [Wei J, et al., Cell 151: 1576-91, 2015]. We found that the addition of lactate to low glucose medium restored Runx2 gene expression to a level similar to that noted with the high glucose medium. However, complete recovery of Sp7 and Tnap mRNA levels was not observed, which may have been caused by promotion of the AMPK-dependent proteasomal degradation of Runx2 due to decreased glucose uptake in low glucose medium. Therefore, it is conceivable that the mRNA expression levels of Sp7 and Tnap, downstream genes of Runx2, were slightly increased by histone lactylation. However, the detailed mechanism is not well understood. These issues are now discussed in the Discussion section of the revised manuscript (page 10-11, lines 225-237).

3-3) This suggestion is greatly appreciated. We have added intracellular lactate graphs (Fig. 2a, 3a), which show a correlation of intracellular lactate concentration with histone lactylation.

Comment #4:

1) In Figure 4, gene expression profiles of Runx2, Sp7 and Tnap were similar to those under oxamate treatment in Figure 2 and were different from those in Figure 3. These discrepant results under different treatments give readers a concern about the validity of the authors’ conclusion. 

2) In addition, knockdown of p300/HAT causes inhibition of histone acetylation and may change expression of various genes aside from change in histone lactylation.

Our reply to comment #4 

4-1) As described in our reply to Comment 3, we consider that the change in Runx2 gene expression caused by addition of lactate to low glucose medium was not caused by an increase in histone lactylation, but rather by other factors.

4-2) As noted by the reviewer, Ep300 siRNA must also be considered to regulate gene expression by histone acetylation. Therefore, we conducted additional experiments with various amounts of protein for PanKac. Ep300 siRNA decreased both histone lactylation and histone acetylation, whereas addition of oxamate did not cause a change in histone acetylation (Fig. 4d, Fig. S1, Fig. S3). On the other hand, oxamate suppressed osteoblast differentiation (Fig. 2b,c). Based on these findings, lactate produced by glucose metabolism may promote osteoblast differentiation through histone lactylation rather than histone acetylation via p300. These findings are presented in the revised Discussion section (page 11, lines 239-245).

Comment #5:

It is worthwhile to refer to and discuss about the paper by Karsenty group (Wei J, et al. Cell 151: 1576-91, 2015) demonstrating that glucose uptake via Runx2-dependent expression of Glut1 is required and that Glut1 and Runx2 crosstalk determines osteoblast differentiation.

Our reply to comment #5

Thank you for this good suggestion. Please see our reply to comment #3-2.

Reviewer 3

The comments regarding our manuscript are greatly appreciated and were very helpful. Please note our responses to the suggestions following.

Comment #1:

Considering that histone lactylation is key to understanding the regulatory mechanism in this article, it is noteworthy that only one result is shown in the bar graph of Figures 1, 2, 3, and 4 (d). It is suggested to include more replicates and display their dispersion as standard deviation (SD) along with statistical analysis. Were both bands considered for relative quantification since two bands are observed? Whatever the case may be, it is recommended to clarify this in the Materials and Methods section.

Our reply to comment #1

Thank you for the good question pointing out these issues. The upper band is approximately 17 kDa and contains histones H2A, H2B and H3, while the lower band is approximately 15kDa and contains histone H4. Based on the reviewer’s comments, we conducted additional experiments (n=3) and created a graph by separating the approximately 17 and 15 kDa bands. Those details are presented in the revised Materials and Methods section (page 14, lines 311-315).

Comment #2:

To demonstrate the role of LDHA and the conversion of pyruvate to lactate, oxamate was used. Was a cell viability analysis performed to rule out possible cytotoxic effects?

Our reply to comment #2

Thank you for this good question. To investigate the cytotoxicity of oxamate, an MTS assay was performed on 72 hours of culture, though no significant change was observed. Those results showed that oxamate (10 mM) had low toxicity (Appendix Fig. 3).

Comment #3:

For the statistical analyses, the Student's t-test was used. However, in Figures 2 and 3, more than two experimental groups were used. In such cases, it is more appropriate to use analysis of variance (ANOVA) and post-hoc multiple comparisons tests.

Our reply to comment #3

Thank you for these helpful comments. According, we performed one-way ANOVA with a post-hoc Tukey test to compare results from more than two groups. A related description has been added to the Statistical analysis section of the revised manuscript (page 14, lines 318-322). 

Comment #4:

Additionally, significant differences are indicated in Figure 3C (Tnap/Gapdh) between low/0 and low/20 mM lactate, but the graph should be modified on its axis to visualize this unclear difference. Furthermore, in this figure, differences between high and low are shown for the Sp7/Gapdh gene, but no expression was detected in one group. It should be clearly stated which groups are being compared in this figure.

Our reply to comment #4

Changed the vertical axis of the graph to logarithmic scale (Fig3c:Tnap/Gapdh).

Based on these helpful suggestions, the graph in Figure 3c showing Sp7/Gapdh was examined again. The low glucose group was ND, so Student’s t-test was used to compare the high and the low glucose medium after adding lactate. A related description has been added to the revised Statistical analysis section (page 14, lines 322-324). In addition, also in Figure 3c (Tnap/Gapdh), the significant difference has been revised because we changed from Student’s t-test to Tukey test results (low glucose medium vs low glucose medium with added lactate).

---

## [Decision Letter · Decision Letter 1]

18 Oct 2023

Lactate-induced histone lactylation by p300 promotes osteoblast differentiation

PONE-D-23-14523R1

Dear Dr. Sasa,

We’re pleased to inform you that your manuscript has been judged scientifically suitable for publication and will be formally accepted for publication once it meets all outstanding technical requirements.

Kind regards,

Atsushi Asakura, Ph.D

Academic Editor

PLOS ONE

Additional Editor Comments (optional):

Reviewers' comments:

Reviewer's Responses to Questions

**Comments to the Author**

1. If the authors have adequately addressed your comments raised in a previous round of review and you feel that this manuscript is now acceptable for publication, you may indicate that here to bypass the “Comments to the Author” section, enter your conflict of interest statement in the “Confidential to Editor” section, and submit your "Accept" recommendation.

Reviewer #1: All comments have been addressed

Reviewer #2: All comments have been addressed

Reviewer #3: All comments have been addressed

2. Is the manuscript technically sound, and do the data support the conclusions?

Reviewer #1: Yes

Reviewer #2: Yes

Reviewer #3: Yes

3. Has the statistical analysis been performed appropriately and rigorously? 

Reviewer #1: Yes

Reviewer #2: Yes

Reviewer #3: Yes

4. Have the authors made all data underlying the findings in their manuscript fully available?

Reviewer #1: Yes

Reviewer #2: Yes

Reviewer #3: Yes

5. Is the manuscript presented in an intelligible fashion and written in standard English?

Reviewer #1: Yes

Reviewer #2: Yes

Reviewer #3: Yes

6. Review Comments to the Author

Reviewer #1: My comments were adequately addressed by the authors. The quality of the manuscript was improved after revision.

Reviewer #2: (No Response)

Reviewer #3: (No Response)

7. PLOS authors have the option to publish the peer review history of their article (what does this mean?). If published, this will include your full peer review and any attached files.

Reviewer #1: **Yes: **Xiangzhen Shen

Reviewer #2: No

Reviewer #3: No

---

## [Editor Report · Acceptance letter]

26 Oct 2023

PONE-D-23-14523R1 

Lactate-induced histone lactylation by p300 promotes osteoblast differentiation 

Dear Dr. Sasa:

I'm pleased to inform you that your manuscript has been deemed suitable for publication in PLOS ONE. Congratulations! Your manuscript is now with our production department. 

Kind regards, 

on behalf of

Dr. Atsushi Asakura 

Academic Editor

PLOS ONE